# Journal article publishing in the social sciences and humanities: A comparison of Web of Science coverage for five European countries

**Michal Petr**[1]*, **Tim C. E. Engels**[2], **Emanuel Kulczycki**[3], **Marta Dušková**[4], **Raf Guns**[2], **Monika Sieberová**[1], **Gunnar Sivertsen**[5]

1 Research Office, Masaryk University, Brno, Czech Republic, 2 Faculty of Social Sciences, Centre for R&D Monitoring (ECOOM), University of Antwerp, Antwerp, Belgium, 3 Scholarly Communication Research Group, Adam Mickiewicz University in Poznań, Poznań, Poland, 4 Slovak Centre of Scientific and Technical Information, Bratislava, Slovak Republic, 5 Nordic Institute for Studies in Innovation, Research and Education, Oslo, Norway

* petr@rect.muni.cz

**Data Availability Statement:** The data underlying this study are publicly available at: https://doi.org/10.5281/zenodo.4060376.

## Abstract

This study compares publication pattern dynamics in the social sciences and humanities in five European countries. Three are Central and Eastern European countries that share a similar cultural and political heritage (the Czech Republic, Slovakia, and Poland). The other two are Flanders (Belgium) and Norway, representing Western Europe and the Nordics, respectively. We analysed 449,409 publications from 2013–2016 and found that, despite persisting differences between the two groups of countries across all disciplines, publication patterns in the Central and Eastern European countries are becoming more similar to those in their Western and Nordic counterparts. Articles from the Central and Eastern European countries are increasingly published in journals indexed in Web of Science and also in journals with the highest citation impacts. There are, however, clear differences between social science and humanities disciplines, which need to be considered in research evaluation and science policy.

## Introduction

From ca the 1980s, bibliometric methods became increasingly anchored in science policy and research evaluation. In this context, bibliometric research refocused from supporting sociological theories to research evaluation topics: science mapping, systemic effects of evaluation procedures, developing new indicators or studying publication patterns [1]. The common attribute of the recent era in bibliometric scholarship is using publication metadata as the primary data source. Major international bibliographic database Web of Science (WoS) has been used in research dealing with publishing and citation characteristics and patterns changes in the social sciences and humanities (SSH), as summarized by Nederhof [2] and recently by Franssen and Wouters [1]. In 2006, Archambault et al. [3] discussed the limits of WoS for any comparative analysis in terms of SSH output due to the language-related database bias. As

**Funding:** The work was supported by the COST Action CA15137 "European Network for Research Evaluation in the Social Sciences and the Humanities" (ENRESSH) through an STSM grant to Michal Petr. The work of Raf Guns and T.C.E. Engels is supported by the Flemish Government through its funding of the Flemish Centre for Research & Development Monitoring (ECOOM), and the work of Gunnar Sivertsen was supported by the Research Council of Norway, Grant 256223 FORINNPOL, the work of Emanuel Kulczycki was supported by the National Science Centre in Poland, Grant UMO-2017/26/E/ HS2/00019. The funders had no role in study design, data collection and analysis, decision to publish, or preparation of the manuscript.

**Competing interests:** The authors have declared that no competing interests exist.

documented by Franssen and Wouters [1], the recent establishment of comprehensive national databases has opened a new direction of bibliometric studies of SSH allowing for a more complete picture of the publication practices by covering more publication types, sources, and research with local relevance than highly selective international databases WoS and Scopus do [4]. However, national databases are relatively new and limit the research by having a different scope and comprehensiveness. In this descriptive study, drawing on the latest bibliometric research approaches and results using national databases as a main data source, we aim to expand knowledge about publication patterns in journal publishing in SSH. More specifically, on an aspect of performance perceived by national evaluation systems in several countries as a sign of good performance, i.e. ranking of journals. We focus on Central and Eastern European (CEE) countries, which comprise a still-neglected region in terms of bibliometrics. Although some research encompassing several CEE countries has been conducted, the overall situation regarding SSH outcome in this region has not been fully grasped. A recent comparative study of eight European countries [5] discovered the changes and differences in publication patterns in terms of the proportion of publication types and publication languages across European countries and disciplines. The authors of the study argued that these differences are often due to the countries' cultural and historical backgrounds. Similarly, Kozak et al. [6] concluded that in terms of international collaboration, number of articles, and citation impact, publication practices and the intensity of change therein differ between individual Eastern European post-communist states, noting that the number of articles had increased the most in the Czech Republic and Poland.

The evaluation systems in the Czech Republic [7] and Poland [8] favour journal articles indexed in Web of Science (WoS) and Scopus over other journal articles, books, and other types of outputs. However, SSH researchers in the Czech Republic still commonly believe that publishing in WoS journals can be at times too challenging for several reasons; whether real or perceived, these reasons include research limited to topics of local relevance, language barriers, and lack of journals in the researchers' fields [9,10]. Some researchers in other CEE countries hold similar opinions [11,12]. The local versus international dilemma also obviously applies to journals' publishing strategies [13,14].

Jurajda et al. [15] analysed the performance of SSH disciplines in post-communist countries through journal-level Article Influence Scores and concluded that performance in these countries lags behind performance in the West. A few years earlier, Vanecek [16] reached a similar conclusion based on his finding that there had been no change in the quality of publications by Czech authors based on the average impact factor (IF) of journals published in all disciplines. Several questions then emerge: are journal-level indicators used in national evaluation systems suitable for studying the developments in SSH publication output or even for evaluating these disciplines at the national level? Furthermore, is the mindset among CEE researchers still defensible when comparing publication patterns in SSH disciplines with patterns in other countries?

In this study, we investigate whether the changes and differences in publication patterns between CEE countries also affect the coverage of SSH articles in WoS and, within WoS, the proportion of articles in Q1 and Q2 journals ranked by IF and subject matter.

We pose the following two research questions:

RQ1: Have SSH publication patterns in CEE countries changed in favour of WoS-indexed articles?

RQ2: Have publication patterns changed in favour of articles published in journals ranked Q1 and Q2 in the Journal Citation Reports (JCR) based on IF?

Alongside each research question, we also ask if there are differences between SSH disciplines. Several previous studies have investigated SSH coverage in major international databases [17–20], but these studies examined only Nordic or Western countries. Here, we will focus on the Czech Republic, Slovakia, and Poland, countries representing the CEE region, because of their similar cultural and political heritages. We will compare them with Flanders (Belgium) and Norway, representing Western and Nordic countries, respectively. The previously mentioned studies by Kozak et al. [6] and Jurajda [15] used WoS only as a data source. Here, we will go beyond WoS coverage and analyse the publication output reported in national databases for all SSH disciplines. Furthermore, we will investigate the journal article share in total peer-reviewed publications registered in national bibliographic databases to obtain a broader picture of publishing practices.

Full coverage of scholarly publication channels in WoS is virtually impossible in SSH disciplines. Publication patterns, channels, and types are much more heterogeneous in SSH disciplines than in science, technology, and medicine (STM). Even though Clarivate Analytics (formerly Thomson Reuters) began indexing more journals published in national languages in 2006, its coverage of SSH publications is still limited [4,18,21]. The Emerging Sources Citation Index (ESCI) was launched in 2015 as a relatively new part of the WoS Core Collection, with backfiles dating to 2005. As of October 2018, this index covered 7,743 journals. The ESCI supplies the Core Collection with SSH journals in local languages and journals of regional scope. The selection process is similar to that for journals indexed in the JCR and in the Arts & Humanities Citation Index (AHCI). Like AHCI publications, ESCI journals do not receive an IF but are included in coverage analysis.

In this study, we consider IF to be an indication of a journal's citation impact. Following this rationale, we take a journal's quartile ranking in the JCR to be an indication of the journal's quality requirements, selectivity, and scholarly impact. Those with higher rankings (Q1 and Q2) are more demanding journals. In this paper, we do not discuss whether IF is an appropriate indicator for evaluating research. We only use it at the journal level, which it was designed for. At the same time, we do not claim that WoS-indexed journals are better than non-WoS-indexed journals. Our selection of indicators reflects the fact that all five countries in our study make use of bibliometric indicators in their national funding and evaluation systems. This common trait in these countries represents just one part of the greater diversity of evaluation systems across Europe [22]. In Poland, journals with IF are given more weight in the Polish Journal Ranking, which is a key element in the national performance-based research funding system (PRFS) [23]. In the past, the Czech PRFS used a formula comprising IF-based journal rankings to calculate publication points. Since 2017, however, calculating the number and proportion of articles in each quartile in journal rankings based on Article Influence Score (AIS) has been one of five modules used in the Czech Republic's PRFS for assessing publication performance in each research field [24]. A similar system is applied in Slovakia [25]. In addition, we would direct the reader to dedicated comparisons of performance-based research funding systems in the EU [26,27].

Although coverage of SSH publications is still limited, which creates unnecessary tensions between SSH disciplines with different degrees of coverage in the databases, the presence of publications in Scopus or WoS has increasingly become a SSH research evaluation criterion [28]. The use of journal-based indicators or journal rankings at the national level can influence the publication behaviour of individuals and institutions [13,29–31]. Vanholsbeeck at al. [32] have demonstrated that even if researchers are critical of how research quality is defined in such evaluation and funding systems, these systems still influence these scholars' publishing priorities and research interests. A previous study focused on Flanders and Norway revealed that the design of evaluation and funding systems affects the intensity of focus on WoS

coverage and WoS-based indicators [18]. The Czech Republic, Poland, Slovakia, Flanders, and Norway have a stronger focus on WoS coverage, and therefore we theorize this focus could influence national-level publication patterns in these countries.

## Data and methods

In this study we use data about peer-reviewed publications from 2013 to 2016 that we collected from national databases in the Czech Republic (RIV), Poland (PBN), Slovakia (CREPČ), Flanders (VABB-SHW), and Norway (NSI). The authors of previous studies have described the need to use national databases as essential data sources for SSH-related analytics, data collection methodologies, definitions of publication types, and inclusion criteria in different countries [5,33,34].

We should note a few important points before proceeding. In the case of Flanders, the VABB-SHW database covers only the Flemish region but not the whole country of Belgium. As for Slovakia, we work with only a 3-year window because data collection started in 2014 (Table 1). For the purposes of this study, we use only data classified in three publications types: a) journal articles, b) monographs and edited books, and c) book chapters and conference proceedings. Details about each country's database and publication types were published in an earlier report [33]. Because national databases evolved in different historical and political settings and are unique in terms of scope, coverage, structure of publication types, and data availability, we must consider the context of data sources when interpreting the results. Differences in the comprehensiveness of national databases, their data structure, scope, and covered period limit the range, in which the data are reliable in mutual comparison. Since we endeavour to study all SSH disciplines, we have adopted the limited timeframe of 2013–2016 for which data are available across all databases, with the exception of Slovak data (2014–2016).

Publications are counted using full counting with the same weight given to all publication types and affiliated countries. Our classification of publications was adopted from the

**Table 1. The total volume of all types of SSH publications in national databases.**

|  | CZE (RIV) | SLO (CREPČ) | POL (PBN) | NOR (NSI) | FLA (VABB-SHW) |
|---|---|---|---|---|---|
| Psychology | 2,099 | n/a | 7,539 | 2,625 | 2,675 |
| Economics and business | 6,770 | 13,973 | 73,119 | 4,105 | 2,864 |
| Educational sciences | 8,260 | 7,004 | 19,285 | 4,037 | 1,186 |
| Sociology | 3,345 | n/a | 12,807 | 2,155 | 2,057 |
| Law | 6,617 | 5,659 | 36,479 | 1,820 | 4,987 |
| Political science | 11,389 | n/a | 14,868 | 2,198 | 1,389 |
| Social and economic geography | 518 | n/a | n/a | 1,754 | 1,026 |
| Media and communication | 917 | n/a | 3,604 | 875 | 770 |
| Other social sciences | 1,262 | 5,737 | 7,960 | 2,808 | 411 |
| History and archaeology | 10,786 | 2,137 | 26,190 | 2,120 | 1,647 |
| Languages and literature | 7,690 | n/a | 47,258 | 3,175 | 3,345 |
| Philosophy, ethics, and religion | 3,981 | n/a | 17,381 | 2,309 | 2,219 |
| Arts | 5,662 | 479 | 5,449 | 1,090 | 1,108 |
| Other humanities | 16 | 11,306 | 5,099 | 734 | 390 |
| **All** | 69,312 | 46,295 | 277,038 | 31,805 | 15,811 |
| **Period** | 2013–2016 | 2014–2016 | 2013–2016 | 2013–2016 | 2013–2016 |

CZE Czech Republic, SLO Slovakia, POL Poland, NOR Norway, FLA Flanders.

Disciplines are ordered as they appear in the Frascati classification scheme.

categories included in the OECD's Frascati Field of Research and Development (FORD) classification scheme [35]. Unfortunately, some disciplines in the Slovak national database, CREPČ, are defined too broadly; thus, they could be classified only as "Other social sciences" or "Other humanities", respectively.

We aggregated the data on two levels:

Level 1: all peer-reviewed publications. To analyse the article share, we used the total counts of all peer-reviewed articles in journals, monographs and edited books, and chapters and conference proceedings collected from national databases, which totalled 449,409 publications.

Level 2: journal articles. To analyse coverage and to link articles with information about journal indexation, we used the full list of journal articles that included basic bibliographic information, FORD classification [29], and digital identifiers (UT WoS, DOI) from every country under analysis. We examined a total of 194,876 articles. For this study, Clarivate Analytics provided us with lists of journals covered by WoS in 2013–2016 that incorporated information about journals' inclusion in WoS citation indexes (SCI-E, SSCI, AHCI, and ESCI) for every year in the studied period. In some cases, these data indicated that a journal was associated with more than one citation index. As we study the coverage and "quality sign" from the perspective of national evaluation systems methodologies, we decided to assign just one index to each journal and used the following decision-making strategy. We prioritised searching in the SCI-E and SSCI (i.e., journals indexed in the JCR), but if a journal was not in the JCR, we searched for it first in the AHCI and then in the ESCI. We also assigned a journal to the AHCI if it was present in both the AHCI and the SCI-E/SSCI but did not have an IF. We also did not duplicate the publications if the journal was present in both SCI-E/SSCI and AHCI. In this case, we treat these journals once as present in JCR (having an IF). As a second step, we assigned a quartile ranking (Q1–Q4) to journals indexed in the JCR. For this purpose, we worked with the quartile ranking valid in the year a given article was published. For journals assigned multiple subject categories, we decided to use the best performing quartile ranking. Finally, we matched journal indexation information with the list of articles containing each journal's ISSN, e-ISSN, title, and other article-level identification (DOI, UT WoS). This procedure allowed us to determine which journals mentioned in the list of articles from national databases are indexed in the year the given articles were published. We considered all articles published in a given year as covered if a journal was present in any WoS citation index in that year. A total of 3.6% of all WoS-indexed articles from national databases were assigned to Book Citation Indexes (BKCI-S, BKCI-SSH), Conference Proceedings Citation Indexes (CPCI-S, CPCI-SSH), and the category "n/a", which stands for WoS-indexed articles in journals indexed in the SCI-E or SSCI but without IF assigned in a given year. For some purposes (e.g. Figs 5 and 6), we merged these articles into the category "other" (BKCI-S, BKCI-SSH, CPCI-S, CPCI-SSH, "n/a").

We conducted the following analyses to answer our research questions. First, we determined the article share. By *article share* we mean the percentage of peer-reviewed journal articles out of the total number of all peer-reviewed SSH publications (articles, books, book chapters, proceedings) registered in national databases. This variable relates only to patterns in the proportion of publication types regardless of indexation in international databases. Second, we determined the percentage of SSH journal articles indexed in WoS out of all articles registered in national databases (coverage). Third, we calculated the proportions of SSH journal articles in WoS citation indexes assigned to the source journals. For those included in the JCR, we further distinguished four quartiles derived from IF-based subject category rankings. In

many countries, it is customary to refer to quartiles in evaluation-related procedures. Quartiles are perceived as a sign of quality. Therefore, in this paper we seek to demonstrate on one hand if coverage is changing and on the other if the perceived quality of the journals in which articles are published is changing. Thus, we focus on calculating the *percentage of articles in Q1 and Q2 journals*. We assume that in SSH fields the first two quartiles of the JCR ranking represent good performance and prestige [28]. In the Results section, we do not describe disciplines where the total number of WoS-indexed articles per country per year does not reach 50 articles because coverage analysis and finer-grained differentiation of citation databases within WoS does not seem justified for such small units. In the Czech Republic, such "small" fields are Law, Social and economic geography, and Media and communication; and in Poland, Social and economic geography, Media and communication, and Arts. In Slovakia, however, only the fields of Economics and business, Educational sciences, and History and archaeology reached the threshold of 50 articles due to the limited classification scheme in place there. In Norway and Flanders, there were no fields that produced fewer than 50 articles per year.

## Results

The results of the two aggregated groups of SSH disciplines are available in the following forms: as the overall article share in the total number of peer-reviewed publications, as the coverage of articles in WoS, and as their distribution in citation indexes in the WoS Core Collection.

### The journal article share in total publications

Out of the five studied countries, Flanders and Norway generally have the highest article share in overall peer-reviewed publications in both the social sciences and the humanities. In the social sciences, the article share in CEE countries does not exceed 50%, while in Flanders and Norway the share is roughly 60–70% (Fig 1). Journal article share is on the rise in the Czech Republic and Norway and is relatively stable in Slovakia. In Poland and Flanders, journal article share decreased; remarkably, the decline in Poland followed a significant gain in 2013 [5]. In the humanities (Fig 2), overall journal article share is lower than in the social sciences, and

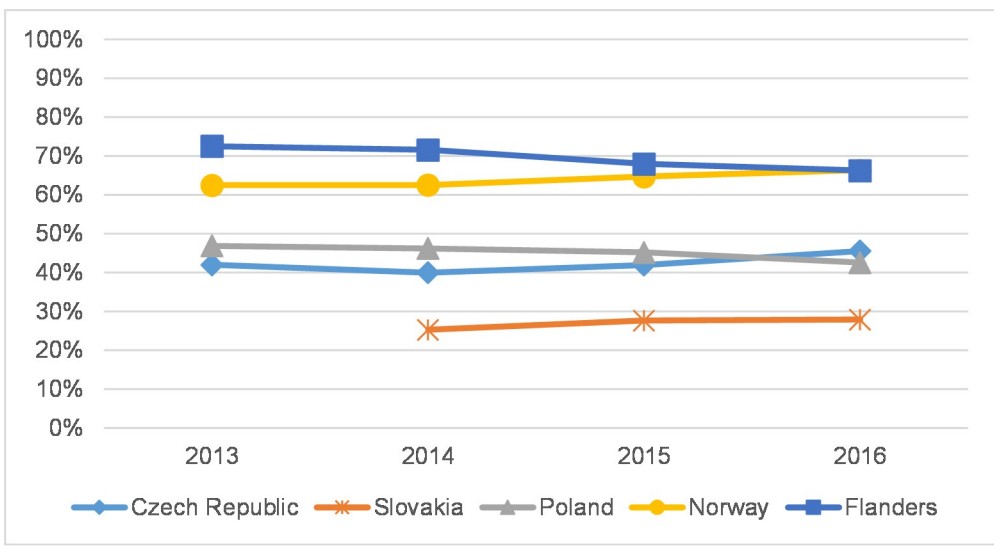

**Fig 1. Article share–social sciences.**

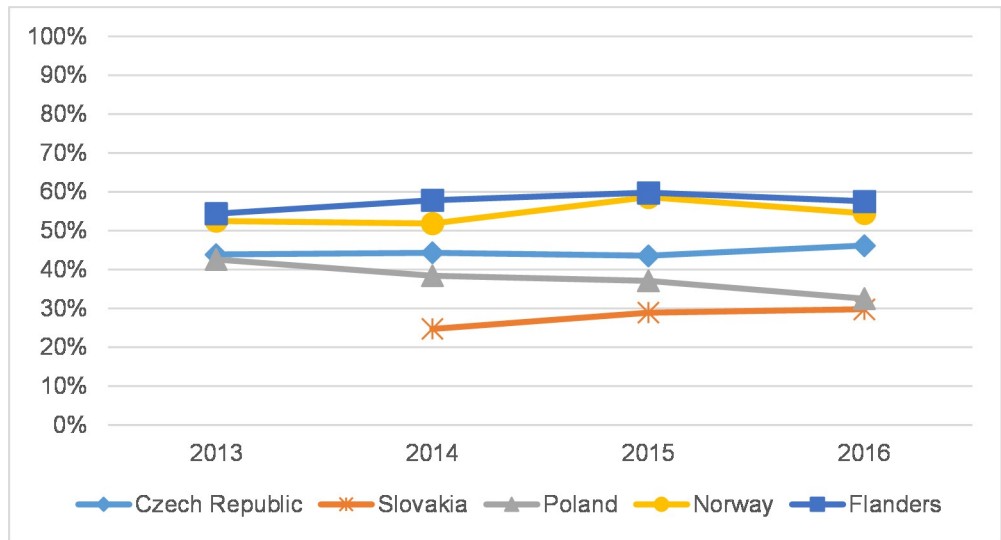

**Fig 2. Article share–humanities.**

the overall differences between countries seem less pronounced. Article share in these fields is mostly stable or slightly increasing. Only in Poland, in nearly all disciplines, is the number of articles as well as the article share stable or on the decline. At the level of individual disciplines, fluctuations seem somewhat greater in the humanities than in the social sciences (S1 and S2 Tables). We observed differences between disciplines rather than differences between countries; in contrast, year-to-year fluctuations are more visible. This analysis serves as an important context for analysing coverage and representation in citation indexes.

## Coverage of journal articles in WoS

Figs 3 and 4 depict developments in the coverage of SSH articles in WoS in 2013–2016. In CEE countries there was a clear increase in the percentage of WoS-covered social sciences articles

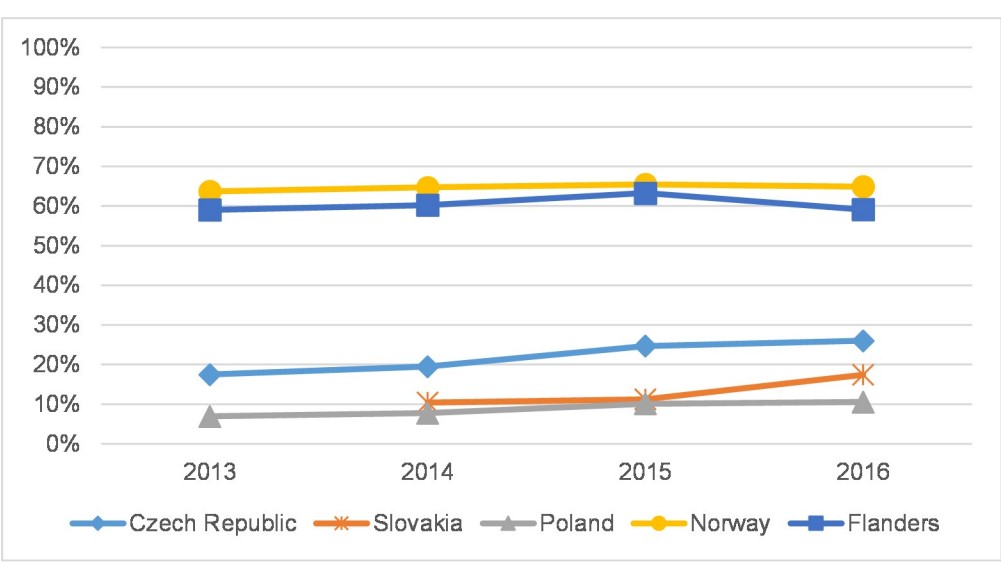

**Fig 3. Coverage of journal articles in WoS–social sciences.**

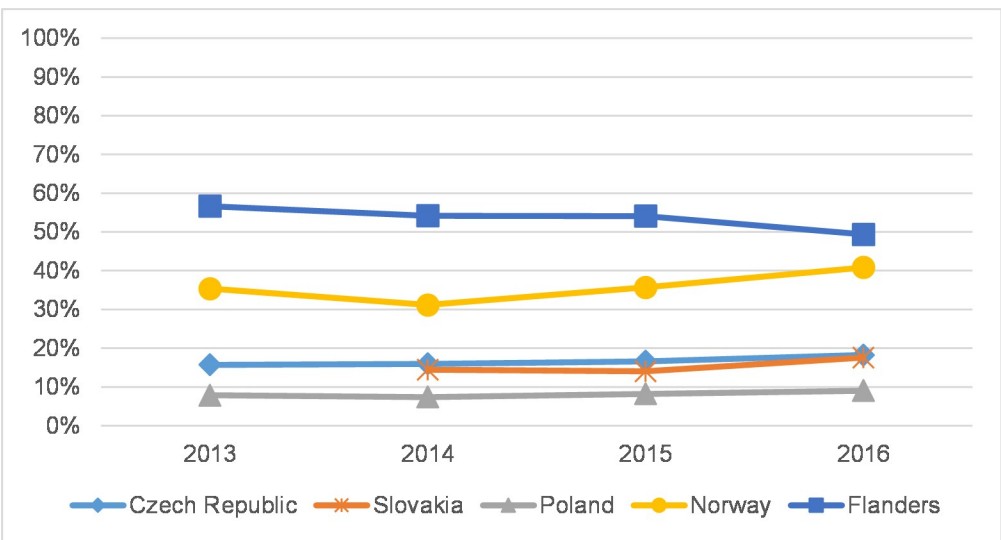

**Fig 4. Coverage of journal articles in WoS–humanities.**

in 2013–2016 (in the Czech Republic from 17.5% to 26.0%, in Slovakia from 10.4% to 17.4%, and in Poland from 6.9% to 10.5%; see Fig 3), albeit absolute values are much lower than in Norway and Flanders, where the proportion of articles in WoS-covered journals is fairly stable at around 65% in Norway and fluctuates around 60% in Flanders. In the humanities (Fig 4), coverage is generally lower, and changes are occurring more swiftly than in the social sciences. Fluctuations are more apparent in the humanities than in the social sciences (S3 and S4 Tables).

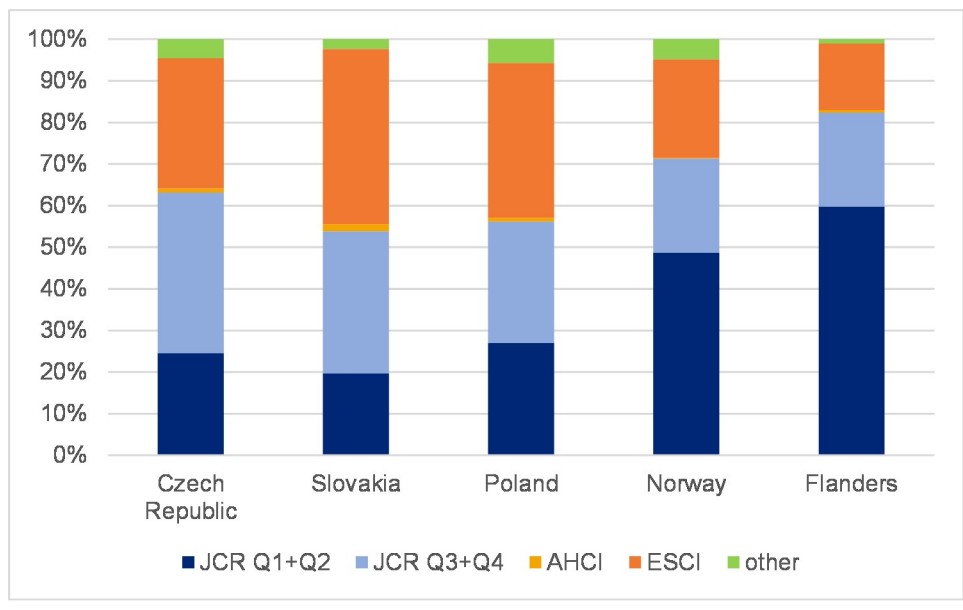

**Fig 5. Proportion of articles in WoS by citation indexes–social sciences.**

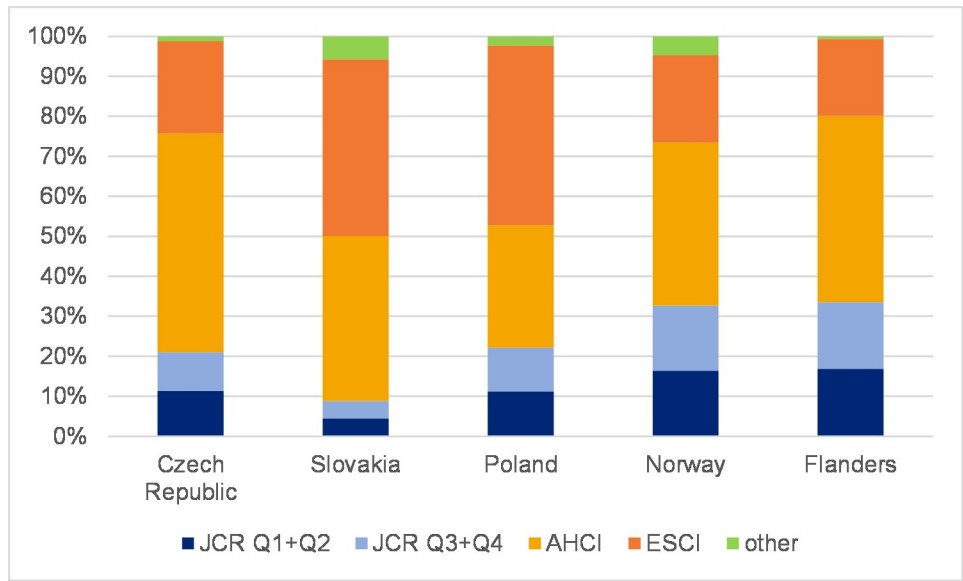

**Fig 6. Proportion of articles in WoS by citation indexes–humanities.**

## The distribution of journal articles in WoS citation indexes

To reveal the extent to which SSH scholars in each country publish in journals in each citation index, Figs 5 and 6 refer to the overall proportion of citation indexes in each country for all the years of the analysis and are divided into a) JCR-indexed journals (with IF)–Q1+Q2; b) JCR-indexed journals (with IF)–Q3+Q4; c) AHCI; d) ESCI; and e) other (BKCI-S, BKCI-SSH, CPCI-S, CPCI-SSH, n/a). Further, Figs 7 and 8 show the percentage of articles published in Q1 and Q2 journals in WoS. In addition, S1–S10 Figs show the distribution of journal articles in citation indexes by country, and S5 Table contains data for each discipline.

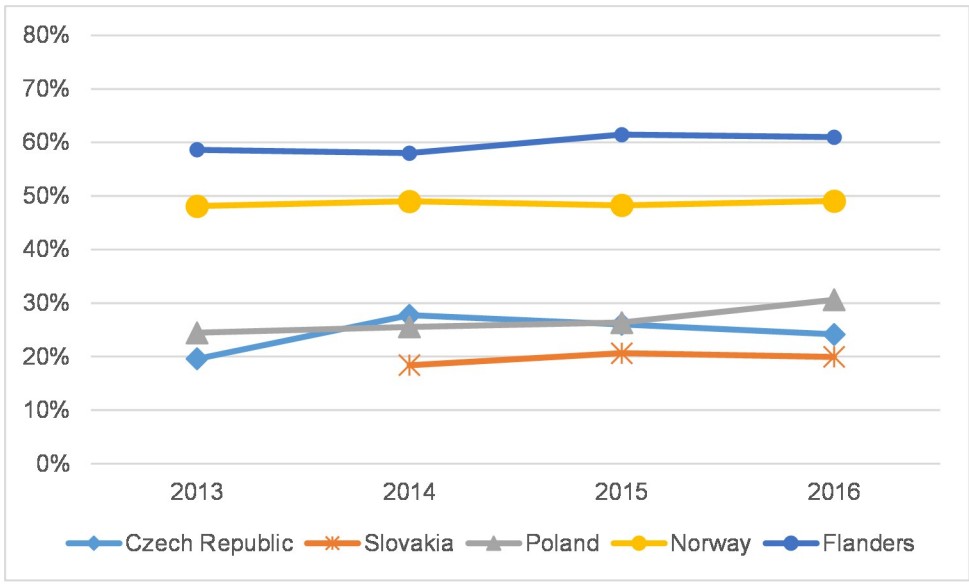

**Fig 7. The percentage of articles in Q1+Q2 journals in WoS–social sciences.**

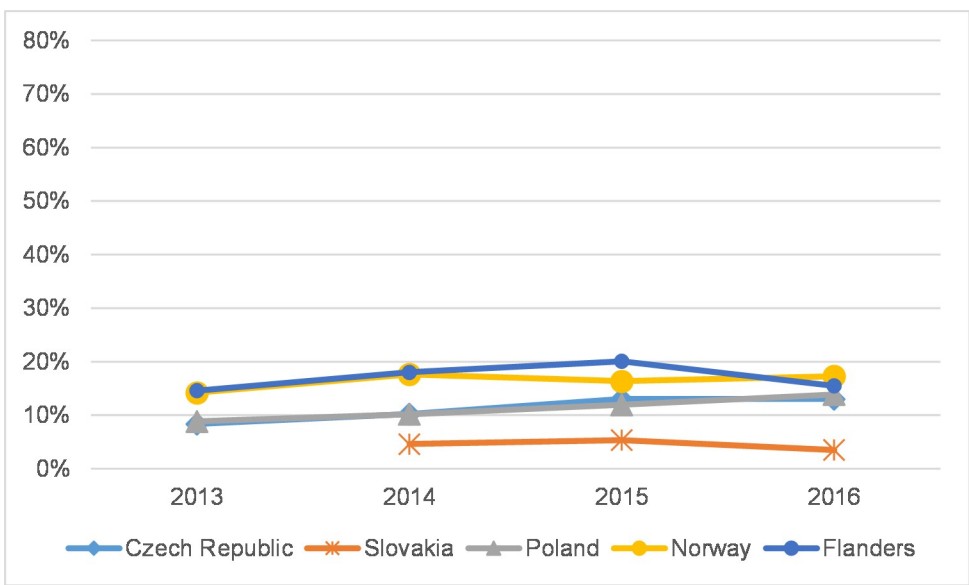

**Fig 8. The percentage of articles in Q1+Q2 journals in WoS–humanities.**

Our analysis revealed that disciplines across the social sciences have similar characteristics in each country. Fig 5 shows that in the Czech Republic the percentage of social science articles published in Q1 and Q2 journals grew overall (24.2% in 2016) as did the percentage of articles in ESCI-indexed journals, whereas the percentage of articles in Q3 and Q4 journals decreased (Table 2). A similar trend can be observed in the social sciences in Slovakia, Poland, and Norway (S5 Table). The data from Flanders indicate a rather stable trend throughout the observed period. The proportion of journals with IF overall and, within this subset of publications, in Q1 and Q2 is, nonetheless, much higher in Flanders and Norway than in CEE countries. A substantial increase in the percentage of articles published in ESCI journals was recorded in Slovakia (from 29.9% in 2014 to 50.3% in 2016) (Table 2); an increase was also apparent in the Czech Republic and Norway (S1–S5 Figs).

Humanities articles indicate a different distribution pattern in citation indexes than social sciences articles. Only moderate changes have occurred in distribution patterns. Fig 6 depicts the percentage of articles in influential journals in each country. The distribution of articles published in all WoS-indexed journals for each country is presented in S6–S10 Figs. The percentage of articles published in journals with IF increased in all countries (S6–S10 Figs) except Slovakia. Flanders also experienced a decrease in 2016. Fig 6 further illustrates the importance of the AHCI for the humanities. One must take into account that performance in the humanities is not measured only by publications in journals with IF, particularly top-tier journals. Table 3 shows that changes in the proportion of articles in Q1 and Q2 journals in the humanities seem less important than the vital presence of articles in AHCI-indexed journals. AHCI-indexed journals article share, however, decreased to the benefit of ESCI- (all countries except Poland) and JCR-indexed (all countries) journals, which is most obvious in Flanders. Slovakia has the lowest share of journal articles with IF (8.9% in 2016), whereas articles in other journals, mostly ESCI-indexed ones, dominate (about 91.1% in 2016, Table 3). Norway is rather stable with some progress towards more articles in journals with IF and more ESCI-indexed articles (S9 Fig).

**Table 2. Distribution of articles in WoS citation indexes–social sciences.**

| | year | JCR Q1+Q2 | | JCR Q3+Q4 | | AHCI | | ESCI | | Other | |
|---|---|---|---|---|---|---|---|---|---|---|---|
| | | # | % | # | % | # | % | # | % | # | % |
| **CZE** | | | | | | | | | | | |
| | **2013** | 150 | 19.6% | 335 | 43.8% | 5 | 0.7% | 222 | 29.0% | 53 | 6.9% |
| | **2014** | 233 | 27.7% | 348 | 41.4% | 7 | 0.8% | 204 | 24.3% | 48 | 5.7% |
| | **2015** | 278 | 26.0% | 395 | 37.0% | 8 | 0.7% | 344 | 32.2% | 43 | 4.0% |
| | **2016** | 276 | 24.1% | 394 | 34.5% | 14 | 1.2% | 425 | 37.2% | 34 | 3.0% |
| **SLO** | | | | | | | | | | | |
| | **2013** | n/a | n/a | n/a | n/a | n/a | n/a | n/a | n/a | n/a | n/a |
| | **2014** | 53 | 18.4% | 121 | 42.0% | 7 | 2.4% | 86 | 29.9% | 21 | 7.3% |
| | **2015** | 66 | 20.6% | 118 | 36.9% | 5 | 1.6% | 123 | 38.4% | 8 | 2.5% |
| | **2016** | 100 | 20.0% | 139 | 27.7% | 6 | 1.2% | 252 | 50.3% | 4 | 0.8% |
| **POL** | | | | | | | | | | | |
| | **2013** | 331 | 24.5% | 476 | 35.2% | 13 | 1.0% | 469 | 34.7% | 64 | 4.7% |
| | **2014** | 416 | 25.6% | 445 | 27.3% | 14 | 0.9% | 641 | 39.4% | 112 | 6.9% |
| | **2015** | 542 | 26.4% | 568 | 27.6% | 16 | 0.8% | 795 | 38.7% | 135 | 6.6% |
| | **2016** | 598 | 30.6% | 549 | 28.1% | 22 | 1.1% | 696 | 35.6% | 88 | 4.5% |
| **NOR** | | | | | | | | | | | |
| | **2013** | 996 | 48.1% | 508 | 24.5% | 4 | 0.2% | 461 | 22.3% | 101 | 4.9% |
| | **2014** | 1,081 | 49.0% | 546 | 24.8% | 0 | 0.0% | 488 | 22.1% | 91 | 4.1% |
| | **2015** | 1,143 | 48.3% | 509 | 21.5% | 0 | 0.0% | 569 | 24.0% | 147 | 6.2% |
| | **2016** | 1,272 | 49.1% | 534 | 20.6% | 5 | 0.2% | 674 | 26.0% | 107 | 4.1% |
| **FLA** | | | | | | | | | | | |
| | **2013** | 886 | 58.7% | 359 | 23.8% | 5 | 0.3% | 215 | 14.2% | 45 | 3.0% |
| | **2014** | 942 | 58.0% | 377 | 23.2% | 8 | 0.5% | 283 | 17.4% | 14 | 0.9% |
| | **2015** | 955 | 61.5% | 311 | 20.0% | 8 | 0.5% | 273 | 17.6% | 7 | 0.5% |
| | **2016** | 1,069 | 61.0% | 405 | 23.1% | 11 | 0.6% | 259 | 14.8% | 9 | 0.5% |

CZE Czech Republic, SLO Slovakia, POL Poland, NOR Norway, FLA Flanders.

## Regional journals in the Czech Republic, Slovakia, and Poland

Because coverage in WoS in 2013–2016 may have increased due to the addition of new regional journals [17], we analysed the proportion of regional journal articles in each CEE country in both the social sciences and the humanities. By the term *regional* we mean journal articles in the national database published in the same country. For Czech and Slovak journal articles we count both Czech and Slovak journals due to the similarity of both languages and the two countries' shared history. Fig 9 shows that although in the Czech Republic coverage increased in both the social sciences and the humanities, the share of articles in regional journals decreased, even though five journals had been recently added (those with more than 10 articles in 2013–2016) to WoS during the observed period. A similar trend can be observed in Slovakia (Fig 10), while Poland is the only country where the article share in regional journals remains stable in the social sciences (Fig 11). In this case, there are eight newly indexed Polish journals in the 4.3% WoS-indexed article share. In the humanities, the article share in Polish journals decreased.

## The aggregated characteristics of countries and SSH disciplines

Our results show a gap between Flanders and Norway on the one hand and CEE countries on the other in all three publication production variables we studied: article share in the total

**Table 3. Distribution of articles in WoS citation indexes–humanities.**

| | year | Q1+Q2 | | Q3+Q4 | | AHCI | | ESCI | | Other | |
|---|---|---|---|---|---|---|---|---|---|---|---|
| CZE | | # | % | # | % | # | % | # | % | # | % |
| | **2013** | 39 | 8.4% | 53 | 11.4% | 225 | 48.4% | 103 | 22.2% | 45 | 9.7% |
| | **2014** | 54 | 10.2% | 41 | 7.8% | 273 | 51.6% | 101 | 19.1% | 60 | 11.3% |
| | **2015** | 69 | 13.0% | 40 | 7.5% | 236 | 44.5% | 123 | 23.2% | 62 | 11.7% |
| | **2016** | 72 | 13.0% | 69 | 12.5% | 225 | 40.7% | 149 | 26.9% | 38 | 6.9% |
| SLO | | | | | | | | | | | |
| | **2013** | n/a | n/a | n/a | n/a | n/a | n/a | n/a | n/a | n/a | n/a |
| | **2014** | 8 | 4.5% | 3 | 1.7% | 81 | 46.3% | 47 | 27.7% | 34 | 19.8% |
| | **2015** | 10 | 5.3% | 10 | 5.3% | 72 | 38.8% | 61 | 32.4% | 33 | 18.1% |
| | **2016** | 8 | 3.4% | 13 | 5.5% | 72 | 33.5% | 124 | 53.0% | 11 | 4.7% |
| POL | | | | | | | | | | | |
| | **2013** | 67 | 8.9% | 65 | 8.6% | 233 | 30.8% | 361 | 47.7% | 31 | 4.1% |
| | **2014** | 76 | 10.2% | 90 | 12.1% | 227 | 30.4% | 327 | 43.8% | 26 | 3.5% |
| | **2015** | 100 | 12.0% | 83 | 9.9% | 235 | 28.1% | 405 | 48.4% | 13 | 1.6% |
| | **2016** | 102 | 13.9% | 96 | 13.1% | 211 | 28.7% | 297 | 40.5% | 28 | 3.8% |
| NOR | | | | | | | | | | | |
| | **2013** | 57 | 14.2% | 63 | 15.7% | 156 | 38.9% | 79 | 19.7% | 46 | 11.5% |
| | **2014** | 70 | 17.6% | 67 | 16.9% | 136 | 34.3% | 88 | 22.2% | 36 | 9.1% |
| | **2015** | 81 | 16.4% | 72 | 14.5% | 169 | 34.1% | 109 | 22.0% | 64 | 12.9% |
| | **2016** | 85 | 17.3% | 88 | 17.9% | 188 | 38.2% | 122 | 24.8% | 9 | 1.8% |
| FLA | | | | | | | | | | | |
| | **2013** | 85 | 14.5% | 96 | 16.3% | 274 | 46.6% | 88 | 15.0% | 45 | 7.7% |
| | **2014** | 106 | 17.9% | 95 | 16.0% | 260 | 43.9% | 97 | 16.4% | 34 | 5.7% |
| | **2015** | 98 | 20.1% | 71 | 14.5% | 176 | 36.1% | 118 | 24.2% | 25 | 5.1% |
| | **2016** | 97 | 15.5% | 118 | 18.8% | 262 | 41.9% | 147 | 23.5% | 2 | 0.3% |

CZE Czech Republic, SLO Slovakia, POL Poland, NOR Norway, FLA Flanders.

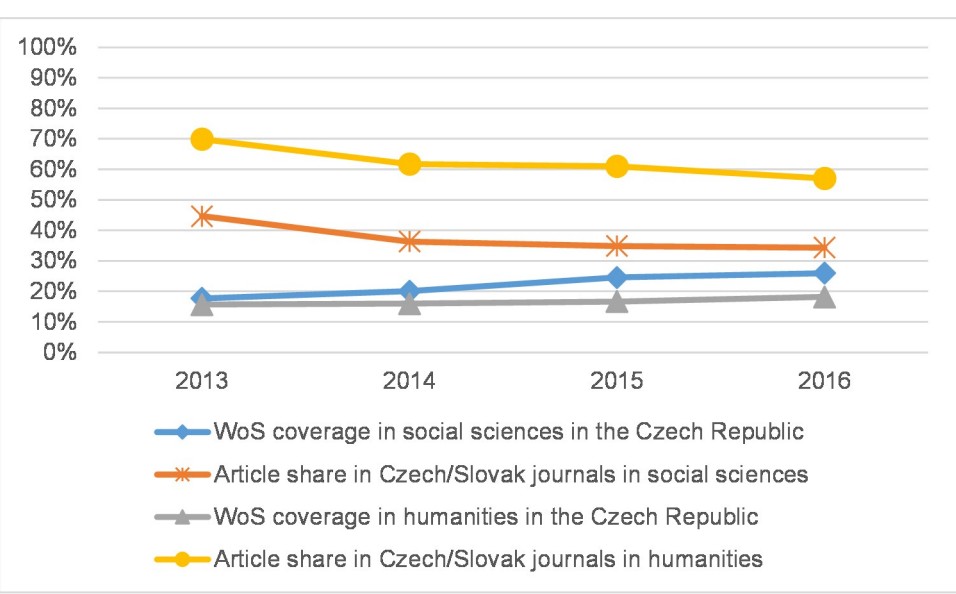

**Fig 9. WoS coverage and regional journal article share in the Czech Republic.**

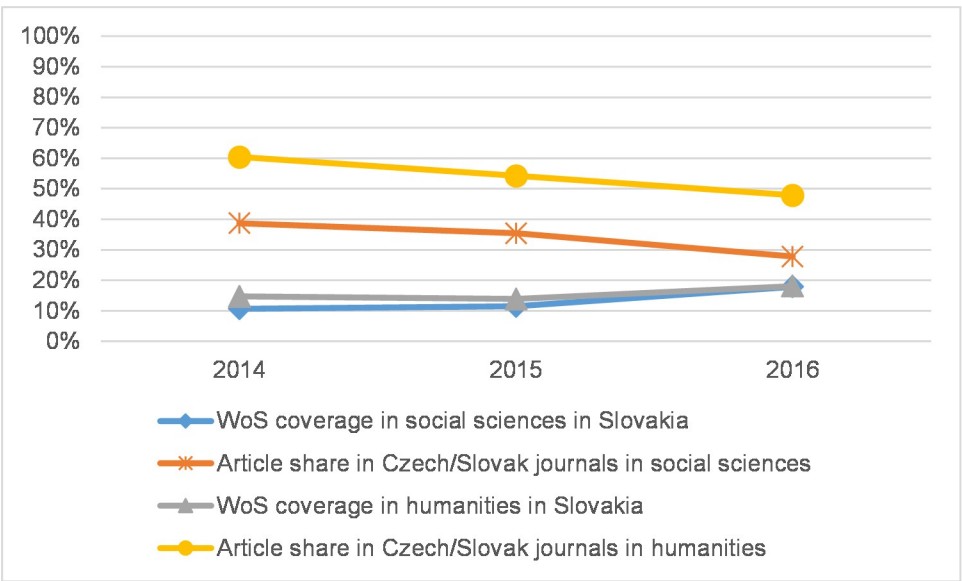

**Fig 10. WoS coverage and regional journal article share in Slovakia.**

production of peer-reviewed publications, coverage of articles in WoS, and percentage of articles in Q1 and Q2 journals. Norway and especially Flanders in most SSH disciplines display higher and relatively stable values in all indicators. Interestingly, in the CEE region, when we compare the Czech Republic and Poland, we discover opposite trends. In the Czech Republic the article share is stable or slightly growing compared to Poland, where the production of articles in all SSH disciplines decreased in 2013–2016 after a significant increase in previous years. The Czech Republic also has a relatively higher article share in most SSH disciplines than other CEE countries, except for in Psychology and Economics and business, where Poland dominates. The Czech Republic also has higher coverage and article representation in Q1 and

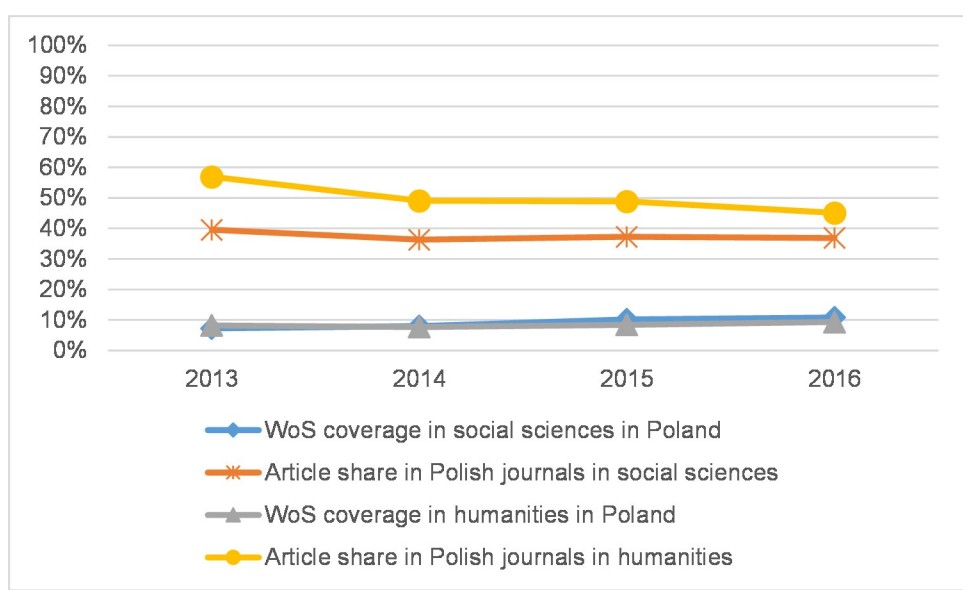

**Fig 11. WoS coverage and regional journal article share in Poland.**

Q2 journals than Poland, at least in fields where articles appear to be a considerable publication channel (Psychology, Economics and business). The situation in Poland is somewhat different. Coverage is not rising (apart from in Psychology) but remains rather stable or has slightly decreased. At the same time though, the percentage of articles in prestigious journals is increasing. Data from Slovakia were analysed less vigorously due to major incomparability of the Slovak national database's classification system and the reduced timeframe of available data. Fluctuations can be attributed to small publication numbers in some disciplines not only in CEE countries, but also in Flanders and Norway in some cases (Law, Media and communication, Arts).

In the field of Psychology, journal articles comprise a usual and stable publication channel (in Norway and Flanders the article share was 80–90%). Most of these articles were published in WoS-indexed journals (in Norway and Flanders coverage in WoS was 80–90%), and a large majority were published in prestigious journals with IF (in Norway and Flanders 60–80% articles were published in Q1+Q2 journals). In this regard, Psychology is rather unusual among SSH disciplines. Elsewhere, certainly in the Czech Republic, journal articles were not always the dominant publication type due to different publication patterns, but, as our data show, this situation is changing. In CEE countries, Psychology has the highest article share and coverage in WoS among all SSH disciplines (Czech Republic 46.7%, Poland 46.2% in 2016; S1 and S3 Tables). In these two countries, publishing trends in Psychology indicate greater coverage in WoS journals and a greater number of articles published in prestigious journals (Czech articles in Q1+Q2 journals rose from 34.1% in 2013 to 42.3% in 2016; Polish articles, from 35.9% to 50.5%). In Poland, Psychology is the only discipline where the coverage of journal articles in WoS exceeded 10% (46.2% in 2016).

The coverage of articles in Economics and business, Political science, and Media and communication shared similar traits in all five countries. In Norway and Flanders, WoS-indexed articles are a stable publication channel, and coverage here was approximately 20% higher than in the Czech Republic, where coverage was increasing. The percentage of WoS-indexed articles in Poland remained below 10%. In this group, Slovak only keeps records for Economics and business, where the data indicate some growth in article share in a limited three-year window (S1 Table). The growing coverage of Czech articles in Economics and business journals (approx. 47% in 2016) was accompanied by a growing percentage of articles in Q1 and Q2 journals (approx. 26.7% in 2016) and a growing percentage of articles in ESCI journals (approx. 30% in 2016, S5 Table). Growing representation in the ESCI could also be seen in Poland and Slovakia, whereas Norway and Flanders published more in JCR journals (S5 Table). The graphs for Media and communication depict a similar profile; the number of WoS-covered articles, however, is too small to conduct further analyses.

In general, the article share in Educational sciences and Law was lower, as was WoS coverage. The countries included in this analysis share similar publication profiles in these disciplines. In both fields, article share was significantly higher in Norway (Educational sciences 54.8%, Law 40% in 2016) and Flanders (Educational sciences 60.9%, Law 63.4% in 2016). In Slovakia and Poland, the article share was rather low (both Educational sciences and Law around 20–30%). In the Czech Republic in Law the article share was relatively higher (46.8% in 2016). Coverage was significantly higher in Norway (Educational sciences 46.4%, Law 25.6% in 2016) and Flanders (Educational sciences 71.4%, Law 15.2% in 2016) than in CEE countries, where coverage in Educational sciences fluctuated between 5 and 15% and coverage in Law remained consistently below 5%. Significantly, CEE countries produced more ESCI-indexed articles than Flanders and Norway. In the Czech Republic and Poland, the share of articles published in influential Q1 and Q2 journals also increased slightly (S5 Table).

Social and economic geography was well represented only in Norway and Flanders but not in Slovakia and Poland, which do entirely not classify this discipline in their databases, and in the Czech Republic, where we found only a few articles.

In the humanities, there were more similarities across disciplines than in the social sciences. For example, data from the fields of History and archaeology, Languages and literature, and Arts demonstrate that publication patterns in the humanities seem to be firmly established. Coverage in WoS was modest in all CEE countries but was greater in Flanders and Norway. Coverage in CEE countries was around 5–15% and was quite stable in 2013–2016. In Norway, however, there was a dynamic change in coverage in the Arts, which jumped from 33.8% in 2013 to 51.6% in 2016. Fluctuations in article share were larger as well. While it is true that differences between countries in article share in the humanities are not as obvious as in the social sciences, the article share in History and archaeology in Flanders and in Arts in Norway was significantly higher than in other countries (around 70% and 60%, respectively).

Publishing patterns in Philosophy, ethics, and religion stand apart from the patterns described above as in this field article share was roughly the same across all countries. Also, the percentage of WoS-covered articles from the Czech Republic (34.2% in 2016) approached coverage in Norway (38.4% in 2016) and Flanders (45.5%, after a decrease in 2016).

## Discussion

This study deepens understanding about current changes in WoS coverage, the increase in three CEE countries in articles published in influential SSH journals, and the general increase in the use of journal articles as a means of communicating SSH research results. Our objectives were to describe the current situation with regard to the coverage of SSH articles in WoS and to identify possible trends towards greater coverage in WoS and towards publishing more in prestigious journals (Q1 and Q2 in ranking by IF) in those articles already covered in WoS.

Kulczycki et al. [5] concluded that publication patterns in 2011–2014 were rather stable in Western and Northern Europe and underwent significant changes in Central and Eastern Europe. Our current findings for 2013–2016 show that this conclusion applies not only to article share, but also to coverage in WoS and representation in citation indexes and, by extension, in prestigious journals based on IF. Publication patterns differed not only between the social sciences and humanities, but also between individual disciplines and between countries within the same discipline.

Some disciplines place more importance than others on journal articles as a publication channel as well as seek to increase the coverage of articles in WoS and the percentage of articles in prestigious journals. This is well documented across all countries in the fields of Psychology and Economics and business. In some countries, other disciplines share these traits, but the national data here vary heavily. In some countries, the article share was significantly higher and may thus be related to locally specific publishing priorities in the field (in Norway in Sociology and Arts, in Flanders in Educational sciences, Law, and Media and communication). Except in Philosophy, ethics, and religion, where the article share was roughly equal across all analysed countries, there were striking differences between the CEE countries and the Western/Nordic countries across all disciplines for all indicators analysed in this paper. That being said, the differences are becoming smaller. The current situation in the Czech Republic, Poland, and Slovakia demonstrates that the number of articles indexed in WoS has clearly increased, and this is also true of a higher percentage of articles in influential journals, though the figures do not reach those of Flanders and Norway. In Flanders, the rapid increase of WoS-indexed journals has been described for the previous period by Engels et al. [17] and Ossenblok et al. [18]. The increase in WoS coverage as described here may be due to some extent to

the addition of new journals, which is especially true in non-English-speaking countries. In this respect, Ossenblok et al. [18] showed, using the examples of Flanders and Norway, that adding a few journals could lead to a substantial difference in terms of WoS coverage for some of the analysed disciplines. Regarding the countries in this research, we hypothesise that the PRFSs in each country could influence journal policy by making indexation in WoS a priority to benefit from favourable publishing conditions for national authors. Macháček and Srholec [36] studied the inclusion of local journals in Scopus in several European countries. They defined a local journal as a journal in which at least 33% of articles are written by domestic authors. As a result, there is a considerably higher number of Scopus-indexed local journals published in CEE countries than in Western countries. In the Czech Republic, this conclusion still applies even when the domestic-author threshold is set at 10%, 33%, or even 66%. Although Macháček and Srholec conducted research on Scopus-indexed journals, their results indicate that specific journal policies were likely stimulated by local PRFSs. In contrast to Macháček and Srholec, we did not carry out an analysis of comparable quality and granularity for WoS; similarly, we did not study the addition of new journals at the level of disciplines. However, the results of our simple comparison of the percentage of articles in regional journals and their coverage in WoS (Figs 9–11) in two research areas (the social sciences and humanities) show that this share decreased even though coverage increased. Therefore, the impact of publishing in regional journals may not have been significant in the observed period.

It may be useful to take into consideration not only differences and developments in article coverage between STM and the SSH, but also between individual SSH disciplines or clusters of disciplines when applying bibliometrics and determining methods for assessing research impact in evaluation systems. Such considerations correspond with De Filippo and Sanz-Casado's [37] findings for three social science disciplines based on an analysis of publication activity, collaboration, impact, and visibility using both traditional and alternative metrics.

Differences in publication patterns are not only related to the publication practices in the given discipline; they are also rooted in differences in scholarly traditions across countries. The gap between CEE countries and Western/Nordic ones is influenced, among other things, by cultural and historical heritage, and more specifically by scholars' ability and willingness to communicate in a foreign language [5]. National PRFSs often provide incentives for adopting different practices; for example, in the Czech Republic, the effect of the PRFS on the strategic behaviour of researchers is well documented [7,12,38,39]. Generally speaking, until 2017 this evaluation system attributed greater weight, quantified as points, to influential WoS- and Scopus-indexed journal articles. Although SSH monographs and other peer-reviewed publications (such as edited volumes and chapters) were taken into account, they were not particularly lucrative [7,16]. Because in the Czech Republic the PRFS had a fundamental impact on the core annual funding of research organisations, it provided incentives for the adaptation of new publishing habits. This was especially true in SSH disciplines, in which producing many non-WoS outcomes or outcomes of mediocre quality was a favourable strategy for ensuring departmental funding [7].

In light of these facts, several research findings deserve to be further commented on. Although publishing in journals is the expected method of communicating results in Economics and business [5], the journal article share in this field is low in the Czech Republic in comparison with Poland. In contrast, we observed a massive number of articles published in a few Czech-based economic journals and an inordinate number of conference proceedings. Neither of these publication types comprise the core of publishing activities in most SSH disciplines [39]. However, the Czech evaluation methodology significantly changed in 2017. By eliminating the straightforward linking of scores ("points") received for individual publications with money, the new evaluation system strives to avoid stimulating unwanted publishing behaviour.

Furthermore, in the case of Poland, the straightforward preference for articles in international journals in national and institutional contexts [23] may have resulted in the increased WoS coverage of articles in many SSH disciplines that this study revealed, although such changes have happened slowly. These observations raise a question: Does change in coverage correspond to researchers' efforts to reflect intradisciplinary changes and/or the push to submit (regional) journals for inclusion in WoS to increase the chances of receiving more money based on the funding mechanisms outlined in national PRFSs? Although from our own experience we know that both options can be valid for some disciplines, our present results (Figs 9–11) show that publishing in domestic journals (regardless of the percentage of domestic authors published in these journals) does not seem to influence increasing coverage. Empirical research is needed to answer this question more properly. However relevant to the results of the present research, studying the formative effects of each country's PRFS on publication strategies was not a goal of this paper. Earlier bibliometric literature pertaining to the effects of PRFSs exist [18,29,31,40,41] and illustrates that PRFSs may influence the publication patterns of scholars in various countries and disciplines.

In this study, we tackled several limitations related to national databases, which were identified and conceptualised in previous studies [34,42,43]. One of the most important limitations of the study lies in cross-country comparison of the data on scientific disciplines determined by different classification methods: cognitive (applied in databases in the Czech Republic, Norway, and Slovakia) or organisational (applied in databases in Poland and Flanders). The analysis and theoretical framework presented by Guns et al. [42] warn about cross-country comparisons such as the present study. According to Guns et al., 73% of Flemish humanities articles, organizationally defined, are published in humanities journals, whereas this ratio is only 59% for the social sciences. Given the objectives of this study, however, we cannot work solely with the cognitive classifications derived from journal classification in WoS, as this could make it difficult to understand how each discipline in each country identifies itself through a set of publications (e.g. an archaeological paper published in an anthropological journal, or an article authored by psychologists published in a neuroscience journal). Thus, one of the possible results of this study is highlighting the ability of disciplines to publish research in outlets that are prestigious or influential (based on IF), even though the journals it is published in may be classified differently in WoS than the articles themselves in the national index. In the Czech system, each publication reported for evaluation is included in the national bibliographic database (RIV) with the cognitive classification determined by the author based on the research topic. But within the evaluation process, the bibliometric analysis is conducted at the level of WoS categories assigned to journals.

Another limitation is rooted in the indexation of some journals in multiple databases, foremost those indexed both in the AHCI and the SCI-E/SSCI-E. This implies a risk of bias regarding the extent to which the SSH are covered in each individual citation index. Presence in multiple citation indexes does not affect article share and overall coverage. In terms of article representation in citation indexes, we decided to link every journal in each year with just one citation index, preferring the SCI-E/SSCI-E over the AHCI, as one of our research questions focuses on changes in the number of articles published in top-tier journals.

In this study, we found differences in the representation of articles in WoS across SSH disciplines and across countries within these disciplines. Research evaluation should respect the diversity of disciplines and not apply mechanisms punishing typical publication patterns in certain fields [28]. English-language journals are overrepresented in WoS. Hence, countries or disciplines publishing more in English are far more likely to be well represented in WoS. This partly explains the lower coverage of humanities publications. In the Czech national evaluation system implemented after 2017, performance in each discipline and in each research

organisation is seen as a mark of "quality" reflecting the distribution of articles in quartiles derived from rankings based on the Article Influence Score. In this respect, the Czech PRFS exposes SSH disciplines to the new challenge of being assessed through publication performance in influential journals. Based on the results of the first two years of the annual evaluation, members of expert panels commenting on the assessment criticised the "apparently" low quality of research in SSH disciplines (unpublished working reports). Previous research also argued that performance in post-communist countries still does not meet the level of that in Western European countries [15,16], but these studies were based solely on the limited view of journal-level indicators. Nevertheless, we should not neglect other publishing activities, for example, AHCI-indexed articles, when interpreting publication performance in the humanities. We argue that the patterns should be seen from a much broader viewpoint that takes into account different contexts and starting points. Assessment results based on a single or inappropriate indicator without understanding the discipline- and country-specific publication patterns and recent developments of disciplines might negatively shape research policies or even the public view of research activities. More specifically, evaluation systems should not punish SSH disciplines on the basis of inappropriate or incomplete measures. Our results offer a basis for a more contextualised interpretation of changes in SSH disciplines.

Differences in publication patterns stem from different national cultural and political backgrounds. Disciplines do not have universal characteristics across all countries. This does not mean, however, that these patterns should be preserved. We do not claim that an article in a WoS-indexed journal is the most demanded publication type in all disciplines, including SSH ones. However, we argue that SSH publishing habits in CEE countries do not need to be maintained based on misplaced or outdated arguments and assumptions. The idea of protecting local excellence, especially in CEE countries, should not be confused with rigidity, avoiding international resources, and the national isolation of SSH disciplines. SSH research in post-communist countries has much to say in international forums, especially the social sciences. Nevertheless, our comparison shows that the potential for having international visibility and impact, understood here as the extent to which SSH disciplines use WoS-indexed journals as a publication channel, can be improved in comparison with Western European and Nordic countries. Locally relevant research published in national languages is especially important for SSH disciplines but preserving local topics and the isolation of SSH disciplines may limit the development of these disciplines in the national and international contexts.

## Conclusions

Our study shows that despite some fluctuations, publication patterns in the social sciences and humanities in CEE countries are changing towards broader representation in WoS, particularly in journals with greater influence in terms of citation impact. Despite the clear differences between the CEE countries studied and the two Western/Nordic countries across all disciplines, publication patterns in CEE countries are becoming more similar to those in Western and Nordic countries, albeit with different intensity. In the Czech Republic, the growing trend in the relative article share in general publication patterns is often related to increased coverage and publication in prestigious journals. In Poland, the article share in all humanities disciplines decreased, but a greater percentage of articles was published in more influential journals. There are nonetheless major potential dissimilarities in the dynamics between countries and individual SSH disciplines. Hence, in bibliometrics, research evaluation, and science policy we suggest taking into account the differences not only between social sciences and humanities but also between disciplines within them.

## Supporting information

**S1 Fig. Proportion of articles in WoS by citation indexes.** Czech Republic–social sciences.
(TIF)

**S2 Fig. Proportion of articles in WoS by citation indexes.** Slovakia–social sciences.
(TIF)

**S3 Fig. Proportion of articles in WoS by citation indexes.** Poland–social sciences.
(TIF)

**S4 Fig. Proportion of articles in WoS by citation indexes.** Norway–social sciences.
(TIF)

**S5 Fig. Proportion of articles in WoS by citation indexes.** Flanders–social sciences.
(TIF)

**S6 Fig. Proportion of articles in WoS by citation indexes.** Czech Republic–humanities.
(TIF)

**S7 Fig. Proportion of articles in WoS by citation indexes.** Slovakia–humanities.
(TIF)

**S8 Fig. Proportion of articles in WoS by citation indexes.** Poland–humanities.
(TIF)

**S9 Fig. Proportion of articles in WoS by citation indexes.** Norway–humanities.
(TIF)

**S10 Fig. Proportion of articles in WoS by citation indexes.** Flanders–humanities.
(TIF)

**S1 Table. Article share–social sciences.**
(DOCX)

**S2 Table. Article share–humanities.**
(DOCX)

**S3 Table. Coverage of journal articles in WoS–social sciences.**
(DOCX)

**S4 Table. Coverage of journal articles in WoS–humanities.**
(DOCX)

**S5 Table. Proportion of articles in WoS by citation indexes.**
(DOCX)

## Acknowledgments

The authors wish to thank Clarivate Analytics for providing the list of journals related to citation indexes in the Core Collection, and reviewers for their valuable comments.

## Author Contributions

**Conceptualization:** Michal Petr.

**Data curation:** Tim C. E. Engels, Emanuel Kulczycki, Marta Dušková, Raf Guns, Monika Sieberová, Gunnar Sivertsen.

**Formal analysis:** Michal Petr, Monika Sieberová.

**Investigation:** Michal Petr, Tim C. E. Engels, Emanuel Kulczycki, Marta Dušková, Raf Guns, Monika Sieberová, Gunnar Sivertsen.

**Methodology:** Michal Petr.

**Writing – original draft:** Michal Petr.

**Writing – review & editing:** Michal Petr, Tim C. E. Engels, Emanuel Kulczycki, Raf Guns, Monika Sieberová, Gunnar Sivertsen.

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
