## [Decision Letter · Decision Letter 0]

22 Jun 2020

PONE-D-20-14121

Coverage of journal articles in social sciences and humanities in Web of Science and their representation in citation indexes: a comparison of five European countries

PLOS ONE

Dear Dr. Petr,

Thank you for submitting your manuscript to PLOS ONE. After careful consideration, we feel that it has merit but does not fully meet PLOS ONE’s publication criteria as it currently stands. Therefore, we invite you to submit a revised version of the manuscript that addresses the points raised during the review process. Please submit your revised manuscript by Aug 06 2020 11:59PM. If you will need more time than this to complete your revisions, please reply to this message or contact the journal office at plosone@plos.org. Please include the following items when submitting your revised manuscript:

We look forward to receiving your revised manuscript.

Kind regards,

Lutz Bornmann

Academic Editor

PLOS ONE

Journal Requirements:

4. Please upload a copy of Figures 10 and 11, to which you refer in your text on pages 17, 23 and 27. If any figure is no longer to be included as part of the submission please remove all reference to it within the text.

5. Please include a copy of Table 1 which you refer to in your text on page 19.

6. Please include captions for your Supporting Information files at the end of your manuscript, and update any in-text citations to match accordingly. Please see our Supporting Information guidelines for more information: http://journals.plos.org/plosone/s/supporting-information

Reviewers' comments:

Reviewer's Responses to Questions

**Comments to the Author**

1. Is the manuscript technically sound, and do the data support the conclusions?

Reviewer #1: Yes

Reviewer #2: Partly

2. Has the statistical analysis been performed appropriately and rigorously? 

Reviewer #1: Yes

Reviewer #2: N/A

3. Have the authors made all data underlying the findings in their manuscript fully available?

Reviewer #1: Yes

Reviewer #2: No

4. Is the manuscript presented in an intelligible fashion and written in standard English?

Reviewer #1: Yes

Reviewer #2: Yes

5. Review Comments to the Author

Reviewer #1: This paper compares comprehensive national records of papers in social sciences and humanities with WoS lists of such papers. Trends over time in Flanders and Norway are compared against Poland, Czech Republic and Slovakia. The paper is solid and I have just a few questions. The text does go on at some length and should be shortened. There were pages and pages of discussion when there really wasn’t that much to discuss, results are just being repeated multiple times. The formatting of the figures made it impossible to consider them in the review. The figures came after the text with no captions associated with them. I don’t know whether this was journal submission requirements or author choice, either way I ended up just ignoring the figures.

On page 4 the way ESCI is described makes one wonder why the journals aren’t just included in WoS. Maybe improve explanation.

I very much liked the interpretation of the IF, very well conceived.

P. 5 “revealing publication patterns of the coverage and ranking of journals in international comparison”

Confusing phrasing, rework the sentence.

p. 5 “A publication in a WoS indexed journal is in most SSH disciplines still seen as a success of its kind and also as an indication of higher quality in publication profile [23]. If this is true, then the ranking of journals might play a role in this process.”

What process? And if what is true? That WoS is seen as higher quality? If it might not be true that it is seen this way, then why did you say it?

Table 1 – numbers in tables should be right justified and ,’s should be used to demarcate 1000’s. And how are the rows in this table ordered? Largest to smallest would be a good order.

“share of articles” sounds like the wrong phrase. It should probably be “article share” Share of articles sounds like the % of articles that are X. You want articles as a % of X.

Table 3 Can you not spell out the country names? And where is the data for Norway?

p. 18 You need to use the field name in the first sentence or two discussing field data in every section. Do not rely on the section heading. Sometime the sentences here can read like they refer to all the data.

p. 23 “ this leads to the same threshold whether” - meaning unclear, rephrase

Reviewer #2: - Data for four (or even three) subsequent single years are insufficient to base conclusions on concerning publication dynamics, let alone trends.

- The analysis of top half IFs vs. bottom half IFs is crude; Subject Category - normalized IFs would be preferable

- Differences between (groups of) countries should be tested statistically where possible

- The decision concerning journals classified in multiple citation indexes to include these in SCI-E and SSCI if possible, and not partly to A&HCI when relevant, might bias findings for Social Sciences as well as for Humanities (p. 8). Preferably, this should be changed, if not, this potential bias should be discussed.

The paper offers an interesting comparison between 3 West Slavic central European countries and two non-English Germanic

ones in North West Europe, concerning their journal article output in SSH. The study of SSH output has a considerable history that, unfortunately,has not been given due consideration. Not much is said about the three elephants in the room: English (the dominant language in scholarly publishing), German (an important language in central and north-western Europe, and Russian the most important Slavic scholarly language in many fields. Especially in the humanities, English is less often used than in most (social) sciences, as opposed to German and Russian. The perhaps most interesting aspect of the paper is the difference in evaluation and reward systems ("PRFS") and in particular the way this relates to adaptive publication strategies of scholars. However, the various PRFSs are described rather summarily for a full appreciation of the findings of the study. Finally, although the importance of books for PRFSs is mentioned, the present study is not able to present much data on these. It is not made clear how well the Book Citation Indexes cover monographs and edited volumes.

Major revisions are recommended to meet the concerns raised above.

Typo's / mistakes: Table 2 should be replaced by Table s2. CZE in Table 2 should be NOR on p.11

6. PLOS authors have the option to publish the peer review history of their article (what does this mean?). If published, this will include your full peer review and any attached files.

Reviewer #1: No

Reviewer #2: Yes: Anton J. Nederhof

---

## [Author Response · Author response to Decision Letter 0]

5 Oct 2020

I also have responded in a more readable way in the attached file - "response to reviewers".

Dear PloS One Editors, 

Foremost, we would like to thank both reviewers for very useful and challenging comments. However, we were struggling with making all requested changes as these sometimes pit against each other. Whereas Reviewer #1 requested shortening the text (which we consider very useful), implementing all changes requested by Reviewer #2 would make the manuscript much longer. Some Reviewer´s #2 suggestions – however undoubtedly overall relevant – goes far beyond the intended focus of this study, changes the scope and would result in the entirely new manuscript. In this respect, we decided giving priority to reducing the text, keeping our original research questions, concept, focus and methodology.

Besides concept-changing requests, we made all requested revisions. Foremost, we reduced and reformulated lengthy Results and Discussion, as Reviewer 1 suggested. At the same time, we added a few paragraphs in the manuscript, discussing Reviewer´s 2 concerns, which we consider very useful and relevant. We believe that we found the right balance and reasonably consolidated the Results and Discussion. We also paid increased attention to improving overall English to make the text clear.

We also uploaded the version with track changes. However, this version only shows our changes in the manuscript and does not contain the changes made during proofreading. Clean version with accepted changes is the final version of revised manuscript.

Below we copied the editor´s review and both Reviewer´s comments. 

Our response is marked with colour and aligned with an increased indent of the paragraph, like this text

***

Editor's comments

We included the title page into the main document as required.

 We uploaded the data in the repository.

4. Please upload a copy of Figures 10 and 11, to which you refer in your text on pages 17, 23 and 27. If any figure is no longer to be included as part of the submission please remove all reference to it within the text.

 We fixed references and figures.

5. Please include a copy of Table 1 which you refer to in your text on page 19.

 Table 1 was at the very beginning of the manuscript. 

6. Please include captions for your Supporting Information files at the end of your manuscript, and update any in-text citations to match accordingly. Please see our Supporting Information guidelines for more information: http://journals.plos.org/plosone/s/supporting-information

 We included captions at the end of the manuscript and checked in-text references.

Reviewers' comments

Reviewer #1

This paper compares comprehensive national records of papers in social sciences and humanities with WoS lists of such papers. Trends over time in Flanders and Norway are compared against Poland, Czech Republic and Slovakia. The paper is solid and I have just a few questions. The text does go on at some length and should be shortened. There were pages and pages of Discussion when there really wasn’t that much to discuss, results are just being repeated multiple times. 

Thank you for this suggestion. We shortened the text, mainly in the Results section. Please, see the revised manuscript.

The formatting of the figures made it impossible to consider them in the review. The figures came after the text with no captions associated with them. I don’t know whether this was journal submission requirements or author choice, either way I ended up just ignoring the figures.

Graphs were uploaded as separate files, according to PLoS One requirements. We fixed all in-text references and included graphs in the text.

On page 4 the way ESCI is described makes one wonder why the journals aren’t just included in WoS. Maybe improve explanation.

P. 5 “revealing publication patterns of the coverage and ranking of journals in international comparison”

Confusing phrasing, rework the sentence.

p. 5 “A publication in a WoS indexed journal is in most SSH disciplines still seen as a success of its kind and also as an indication of higher quality in publication profile [23]. If this is true, then the ranking of journals might play a role in this process.”

What process? And if what is true? That WoS is seen as higher quality? If it might not be true that it is seen this way, then why did you say it?

We looked at how the Introduction could be expressed more clearly, given these questions from Reviewer #1. Finally, we edited the Introduction from its start and improved the explanations.

Table 1 – numbers in tables should be right justified and ,’s should be used to demarcate 1000’s. And how are the rows in this table ordered? Largest to smallest would be a good order.

We updated all Tables as suggested and added following sentence under the Table 1: “Disciplines are ordered as they appear in the Frascati classification scheme.”

“share of articles” sounds like the wrong phrase. It should probably be “article share” Share of articles sounds like the % of articles that are X. You want articles as a % of X.

Thank you. With our proofreader, we rephrased “share of articles” to “article share” and improved overall English.

Table 3 Can you not spell out the country names? And where is the data for Norway?

We added the country names legend below tables. The data from Norway was present – there was a mistake in splitting the table in two pages. We fixed this.

p. 18 You need to use the field name in the first sentence or two discussing field data in every section. Do not rely on the section heading. Sometime the sentences here can read like they refer to all the data.

Concerning the very first comment on lengthy Results, we have reformulated the whole section. Namely, we deleted headings, reduced the text describing each discipline and substituted with the description of findings in publication patterns. Please, see the manuscript.

p. 23 “ this leads to the same threshold whether” - meaning unclear, rephrase

We rephrased an unclear sentence.

Reviewer #2

- Data for four (or even three) subsequent single years are insufficient to base conclusions on concerning publication dynamics, let alone trends.

National databases are heterogeneous as for the scope, coverage, structure of publication types and availability of the data. Databases are also unique from the perspective of historical and political background. From that point of view, comparing data from databases with unique data structure and working conditions is highly problematic. We decided to analyze data for the timeframe 2013–2016 because this was the timeframe for which the data was available across all databases. In some databases, data is collected retrospectively with some delay. Expanding is problematic when we need to align all countries. 

In this study, the data from national databases needed were post-processed to be mutually comparable and to be linked to the information about indexing each journal in each citation index in each year of the study. This procedure is only partly doable automatically. A considerable part of the work needs to be done manually (mostly linking ISSNs and titles to WoS journal lists and verification of the coverage of journals in given years). Therefore, this timeframe is, to a certain extent balanced with great accuracy of the data and with a couple of months (sic!) of data cleaning in background.

Following reviewers recommendations, we achieved to add the year 2016 for Flanders data to align the timeframe with other countries. Slovakia has no older data in the database than 2014. Even when showing 4-years data, we can see overall trends and differences between countries at least in disciplines with a sufficient number of publications. Even in the pre-revised version of the manuscript, we did not base conclusions in disciplines with small publication base and low coverage. 

We have added the following explanation in the Data and Methods section: “National databases are unique as for the scope, coverage, structure of publication types and availability of the data. Databases have also different historical and political background. From that point of view, comparing data from databases is highly problematic. Since we endeavour to study all SSH disciplines, we accepted the limited timeframe 2013–2016 for which the data was available across all databases.”

- The analysis of top half IFs vs. bottom half IFs is crude; Subject Category - normalized IFs would be preferable

Since quartiles are calculated on the basis of the IF ranking within each WoS Category, they already form a crude subject normalization. We use quartiles because it is an understandable way how to show the position of the journal in the WoS category. Also, in many countries, it is customary to refer to quartiles in evaluation-related procedures. Quartiles are here perceived as “quality” sign. Therefore, the idea of this article was to compare the proportions of quartile zones across countries and to show, on the one hand, if the coverage is changing and on the other hand if the perceived “quality” of journals is changing. We explained this directly in the Introduction and also in Data and Methods section. We are of the opinion that IF or any other finer-grained metric would be kind of different information and not so meaningful for the purpose of the study. 

We have rephrased the Introduction as follows: 

“In this study, we consider IF to be an indication of a journal’s citation impact. Following this rationale, we take a journal’s quartile ranking in the JCR to be an indication of the journal’s quality requirements, selectivity, and scholarly impact. Those with higher rankings (Q1 and Q2) are more demanding journals. In this paper, we do not discuss whether IF is an appropriate indicator for evaluating research. We only use it at the journal level, which it was designed for. At the same time, we do not claim that WoS-indexed journals are better than non-WoS-indexed journals. Our selection of indicators reflects the fact that all five countries in our study make use of bibliometric indicators in their national funding and evaluation systems. This common trait in these countries represents just one part of the greater diversity of evaluation systems across Europe.”

We have added the following part to the Data and methods section:

“In many countries, it is customary to refer to quartiles in evaluation-related procedures. Quartiles are perceived as a sign of quality. Therefore, in this paper we seek to demonstrate on one hand if coverage is changing and on the other if the perceived quality of the journals in which articles are published is changing.”

- Differences between (groups of) countries should be tested statistically where possible

Since we work with simple descriptive statistics, we do not consider statistical testing as necessary in this specific situation. It is not clear, what is a benefit of statistical testing for the present study when we show the simple article counts and shares without using more dimensions in one figure.

- The decision concerning journals classified in multiple citation indexes to include these in SCI-E and SSCI if possible, and not partly to A&HCI when relevant, might bias findings for Social Sciences as well as for Humanities (p. 8). Preferably, this should be changed, if not, this potential bias should be discussed.

For the article share and coverage the presence in multiple citation indexes does not play a role. For the representation in citation indexes we counted only one citation index with the decision strategy explained in the manuscript – priority was given to JCR indexation (with IF), then we searched in AHCI and finally in ESCI. The explanation for this decision was the point of view of national evaluation systems and the perception of journal “prestige” or “quality”. 

We looked at the extent to which journals are indexed both in AHCI and SCI-E or SSCI and added the text to the Data and methods section and found that there are 4.4 % articles with multiple assignments to AHCI and SCI-E/SSCI of the total number of WoS-indexed articles. Then there are 27.6 % articles with multiple assignments to AHCI and SCI-E/SSCI out of all AHCI-indexed articles. It is a very interesting phenomenon; however, it goes beyond the extent of this study. Changing the counting method in the way the Reviewer suggests would cause enormous changes in the master dataset, including new time-consuming cleaning and completing the data. Looking at journals with multiple citation index assigned, we revealed many specific situations, deserving manual decision. Then, the deeper conceptualization is needed, what does it mean for the performance in SSH. Although the representation of these journals with multiple indexes is remarkable, we think that treating multiple indexes as separate values at the level of the journal would cause potential inaccuracy and would be of rather low added value to this particular research questions. 

We added the following sentence in the Data and Methods section.

“We also assigned a journal to the AHCI if it was present in both the AHCI and the SCI-E/SSCI but did not have an IF.”

Then we discussed this as a limitation in the Discussion as follows:

“Another limitation is rooted in the indexation of some journals in multiple databases, foremost those indexed both in the AHCI and the SCI-E / SSCI-E. This implies a risk of bias regarding the extent to which the SSH are covered in each individual citation index. Presence in multiple citation indexes does not affect article share and overall coverage. In terms of article representation in citation indexes, we decided to link every journal in each year with just one citation index, preferring the SCI-E/SSCI-E over the AHCI, as one of our research questions focuses on changes in the number of articles published in top-tier journals.”

The paper offers an interesting comparison between 3 West Slavic central European countries and two non-English Germanic ones in North West Europe, concerning their journal article output in SSH. The study of SSH output has a considerable history that, unfortunately,has not been given due consideration. Not much is said about the three elephants in the room: English (the dominant language in scholarly publishing), German (an important language in central and north-western Europe, and Russian the most important Slavic scholarly language in many fields. Especially in the humanities, English is less often used than in most (social) sciences, as opposed to German and Russian. 

Thank you for this suggestion. We have added the following part to the Discussion:

“English-language journals are overrepresented in WoS. Hence, countries or disciplines publishing more in English are far more likely to be well represented in WoS. This partly explains the lower coverage of humanities publications.”

Kulczycki et al. (2020), https://asistdl.onlinelibrary.wiley.com/doi/full/10.1002/asi.24336, have found that the importance of Russian and German is fairly limited in journal publishing in CEE countries nowadays, accounting for, respectively, 0.4 to 1.0% and 1.7 to 2.3% of SSH articles. In other words, both languages are historically important in some of the countries under scrutiny but no longer play a major role. For this reason, we don't specifically mention them in the article

The perhaps most interesting aspect of the paper is the difference in evaluation and reward systems ("PRFS") and in particular the way this relates to adaptive publication strategies of scholars. However, the various PRFSs are described rather summarily for a full appreciation of the findings of the study. 

Thank you for this relevant suggestion. This is an aspect that we are usually intensely interested in as well. However, studying relationships between PRFS and publication strategies directly in the manuscript was not the goal of this paper. Expectedly, it would take a lot of space – especially when describing the details how publications are discounted in those models – whereas Reviewer #1 asked for reducing the text length. We edited the Introduction and added references to recently published research as follows rather than describe each PRFS in detail.

Debackere, K., Arnold, E., Sivertsen, G., Spaapen, J., & Sturn, D. (2018). Performance-Based Funding of University Research. Publications Office of the European Union. https://doi.org/10.2777/644014.

Zacharewicz, T., Lepori, B., Reale, E., & Jonkers, K. (2018). Performance-based research funding in EU Member States—A comparative assessment. Science and Public Policy. https://doi.org/10.1093/scipol/scy041

Finally, although the importance of books for PRFSs is mentioned, the present study is not able to present much data on these. It is not made clear how well the Book Citation Indexes cover monographs and edited volumes.

We agree that books are crucial for PRFSs too. We did not provide any data on this because the main point of the paper was to study the article share, coverage and presence in WoS CC citation indexes with the finer-grained distinction of quartiles within JCR across all SSH disciplines in Frascati classification. It is not possible to cover all topics related to all publication patterns in SSH. The coverage of BCI is very limited in the SSH and is documented by Aksnes and Sivertsen (2019), which is also referenced in the manuscript. 

Major revisions are recommended to meet the concerns raised above.

Typo's / mistakes: Table 2 should be replaced by Table s2. CZE in Table 2 should be NOR on p.11.

We fixed these mistakes.

---

## [Decision Letter · Decision Letter 1]

17 Nov 2020

PONE-D-20-14121R1

Journal article publishing in the social sciences and humanities: a comparison of Web of Science coverage for five European countries

PLOS ONE

Dear Dr. Petr,

Thank you for submitting your manuscript to PLOS ONE. After careful consideration, we feel that it has merit but does not fully meet PLOS ONE’s publication criteria as it currently stands. Therefore, we invite you to submit a revised version of the manuscript that addresses the points raised during the review process. One reviewer is still very critical and recommended to reject the manuscript. Thus, you should address the reviewer's points very carefully and comprehensively. Please submit your revised manuscript by Jan 01 2021 11:59PM. If you will need more time than this to complete your revisions, please reply to this message or contact the journal office at plosone@plos.org. Please include the following items when submitting your revised manuscript:

We look forward to receiving your revised manuscript.

Kind regards,

Lutz Bornmann

Academic Editor

PLOS ONE

Reviewers' comments:

Reviewer's Responses to Questions

**Comments to the Author**

1. If the authors have adequately addressed your comments raised in a previous round of review and you feel that this manuscript is now acceptable for publication, you may indicate that here to bypass the “Comments to the Author” section, enter your conflict of interest statement in the “Confidential to Editor” section, and submit your "Accept" recommendation.

Reviewer #1: All comments have been addressed

Reviewer #2: (No Response)

2. Is the manuscript technically sound, and do the data support the conclusions?

Reviewer #1: Yes

Reviewer #2: No

3. Has the statistical analysis been performed appropriately and rigorously? 

Reviewer #1: N/A

Reviewer #2: No

4. Have the authors made all data underlying the findings in their manuscript fully available?

Reviewer #1: No

Reviewer #2: Yes

5. Is the manuscript presented in an intelligible fashion and written in standard English?

Reviewer #1: Yes

Reviewer #2: Yes

6. Review Comments to the Author

Reviewer #1: (No Response)

Reviewer #2: It is laudable that the authors have provided additional data for Flanders for 2016, but a four-year period is still way to short to detect a trend. It might be possible to test for differences between, for instance, early and late years, but unfortunately, the authors decline to do this. This makes the paper less suitable for PLOS ONE.

A minor point is that the paper still does not pay proper attention to earlier studies dealing with publication changes in SSH.

Concerning the use of quartiles to measure journal impact: a small increase in the IF of a journal may raise it to another quartile; worse, a journal's decrease in IF may be less than that of one or more journals originally ranked in a higher quartile, leading to its promotion to a higher quartile. Using more fine-grained impact measures helps to detect and prevent such statistical problems. This point has not been addressed properly in the revised paper.

The paper's choice to not include journals with an IF partly to AHCI when these are both indexed in AHCI and other citation indexes considerably limits the paper's relevance for the humanities, as does, possibly to a lesser extent, the exclusion of BCI items.

Regrettably, the effect of changes in PRFS (reward and evaluation systems) on changes in publication patterns is still not more fully addressed. This distracts from the potential value of the paper.

Although the revised paper deals satisfactory with a few points have been raised in the original comments, it fails to address several important points.

7. PLOS authors have the option to publish the peer review history of their article (what does this mean?). If published, this will include your full peer review and any attached files.

Reviewer #1: No

Reviewer #2: **Yes: **Anton J. Nederhof

---

## [Author Response · Author response to Decision Letter 1]

16 Feb 2021

Revised cover letter to: Petr, M et al., Journal article publishing in the social sciences and humanities: a comparison of Web of Science coverage for five European countries.

Dear PloS One Editors, 

We are thankful for the Reviewer’s #2 comments that inspired us to make some more revisions and explain some critical points in the manuscript. We also added some extra references, addressing significant topics suggested by Reviewer’s #2. We tried to reflect the Reviewer‘s #2 concerns; however, we still feel that making all changes requested by Reviewer #2 would change the article's scope entirely. For instance, reviewer #2 asked addressing the effects of changes in PRFS on changes in publication patterns. We feel that addressing this point would make this text on the one hand too extensive in the scope and length, on the other hand too general to address these points to a deserved extent. Further, Reviewer #2 requested replacing the IF quartiles with more finer-grained impact measure. We want to remark that using IF quartiles in our article is entirely intentional as several national evaluations use this exact level for assessing performance. However, we addressed several topics in the article, provided some extra explanations and added references to document the Reviewer‘s #2 points.

Reviewer #2 finally says: „Although the revised paper deals satisfactory with a few points have been raised in the original comments, it fails to address several important points.” 

In this article, we do not aim for comprehensiveness; the added value lies mainly in comparing the data from national databases from the perspective of the perception of publication performance by evaluation systems. We do not argue with the overall relevance and importance of recommendations that we did not implement. However, we still see that some implementations (mentioned above) are far beyond the paper's current scope and would significantly change it. Therefore, we prefer to focus on the topics we set up at the beginning of the research, as stated in the article. 

With full respect to the experience of Reviewer #2, we would like to thank you for these beneficial recommendations, which we will be happy to address in further research projects. We kindly ask the Editor to reflect in his decision our argumentation that we carefully considered and still see as reasonable. 

Below we copied the Reviewer´s #2 comments. 

We also replaced the figures with PACE-fixed, as requested by the Editor. 

Reviewers' comments

Reviewer #2: It is laudable that the authors have provided additional data for Flanders for 2016, but a four-year period is still way too short to detect a trend. It might be possible to test for differences between, for instance, early and late years, but unfortunately, the authors decline to do this. This makes the paper less suitable for PLOS ONE.

A minor point is that the paper still does not pay proper attention to earlier studies dealing with publication changes in SSH.

Response: National bibliographic databases are relatively new data sources, and we benefit from the fact that they cover full national output. WoS has been used as the main data source in all periods of bibliometric research on publication patterns, but national databases are used nowadays as an emerging data source with a great potential to analyse publication patterns beyond WoS. However, national bibliographic databases have limitations, such as differences in comprehensiveness, data structure, scope, covered period etc. Limitations having effect e.g. in the period analyzed in our article, pin the territory where our data can be reliable in mutual comparison. Expanding this period is problematic. Using just Web of Science data would make the work several times easier and allow to extend the timeframe of the dataset easily, but we decided to strengthen the focus on referencing the relevant literature on this and on describing these differences. We added the following text and a couple of relevant references to the very beginning of the Introduction: 

From ca the 1980s, bibliometric methods became increasingly anchored in science policy and research evaluation. In this context, bibliometric research refocused from supporting sociological theories to research evaluation topics: science mapping, systemic effects of evaluation procedures, developing new indicators or studying publication patterns [1]. The common attribute of the recent era in bibliometric scholarship is using publication metadata as the primary data source. Major international bibliographic database Web of Science (WoS) has been used in research dealing with publishing and citation characteristics and patterns changes in the social sciences and humanities (SSH), as summarized by Nederhof [2] and recently by Franssen and Wouters [1]. In 2006, Archambault et al. [3] discussed the limits of WoS for any comparative analysis in terms of SSH output due to the language-related database bias. As documented by Franssen and Wouters [1], the recent establishment of comprehensive national databases has opened a new direction of bibliometric studies of SSH allowing for a complete picture of the publication practices by covering more publication types, sources, and research with local relevance than highly selective international databases WoS and Scopus do [4]. However, national databases are relatively new and limit the research by having a different scope and comprehensiveness. In this descriptive study, drawing on the latest bibliometric research approaches and results using national databases as a main data source, we aim to expand knowledge about publication patterns in journal publishing in SSH. More specifically, on an aspect of performance perceived by national evaluation systems in several countries as a sign of good performance, i.e. ranking of journals. [1] Franssen T, Wouters P. Science and its significant other: Representing the humanities in bibliometric scholarship. Journal of the Association for Information Science and Technology. 2019;70:1124-1137. doi: 10.1002/asi.24206.

[2] Nederhof A. Bibliometric monitoring of research performance in the Social Sciences and the Humanities: A Review. Scientometrics. 2006;66:81–100. doi: 10.1007/s11192-006-0007-2. 

[3] Archambault E, Vignola‐Gagne E, Cote G, Lariviere V, Gingras Y. Benchmarking scientific output in the social sciences and humanities: The limits of existing databases. Scientometrics. 2006;68(3):329–342. doi: 10.1007/s11192-006-0115-z. 

As for the trends, we describe the developments in the results section of the manuscript but conclude with more general ideas not entirely dependent on the trends: comparison of countries, main differences and relevant background. 

Reviewer #2: Concerning the use of quartiles to measure journal impact: a small increase in the IF of a journal may raise it to another quartile; worse, a journal's decrease in IF may be less than that of one or more journals originally ranked in a higher quartile, leading to its promotion to a higher quartile. Using more fine-grained impact measures helps to detect and prevent such statistical problems. This point has not been addressed properly in the revised paper.

Response: Although we fully agree with the above-mentioned behaviour of the indicators, we would like to point out that it was not the paper's aim to evaluate journals, assess disciplines, or build a comprehensive picture of SSH. We conducted a descriptive analysis of the state of affairs, expanding the current picture of SSH using national databases as a data source. Our goal is to detect publication patterns in the specific area that is used for evaluating research in several national evaluation systems. Therefore, dealing with small changes to IF influencing the journal’s quartile is not entirely relevant if researchers perceive Q1 journals to be somehow special. For instance, in the Czech Republic, the evaluation methodology ranks journals in IF quartiles strictly due to journals´ IF in the year of publication without paying attention to the influence of small differences in the IF value on the line between two quartiles. In the present article, we show the distribution of journals in citation indexes and more specifically in IF quartiles in the way the national systems use quartiles for evaluating research. We translated this logic with all its imperfections of approximation methods of the national evaluation. We believe that we explained this intention satisfactory in the manuscript.

Reviewer #2: The paper's choice to not include journals with an IF partly to AHCI when these are both indexed in AHCI, and other citation indexes considerably limits the paper's relevance for the humanities, as does, possibly to a lesser extent, the exclusion of BCI items.

Response: The paper's main aim is to analyze the coverage of articles, which is clearly stated in the title and the manuscript. The decision not to duplicate items in each country´s dataset is, as we believe, suitable for the aim we established (see also above). We consider the decision strategy in deduplication of more citation indexes assigned to one journal appropriate to determine the discipline's performance aligned to the method of evaluation systems. Including journals partly in AHCI when these are also indexed in JCR is also possible and relevant for some cases. Nevertheless, we believe that humanities are not reduced in the results in any manner, as we show these articles preferably in the JCR-indexed category in accordance with the assumption that this category performs better in the evaluation system. Also, for the comparison of different countries from the point of view of the presence of articles in JCR-indexed journals, showing these duplicates in AHCI does not play a role 

Reviewer #2: Regrettably, the effect of changes in PRFS (reward and evaluation systems) on changes in publication patterns is still not more fully addressed. This distracts from the potential value of the paper. 

Response: We agree that analysis of the effects of PRFS’s is an important topic deserving of extensive discussion but argue that including such an analysis would need – a lot of explanation and would make the article lengthy. Instead of this, we added several works (references) in the Discussion (p. 27) that have earlier studied the PRFS’s effects (and demonstrated how complicated it is to trace such effects), some of which also brought some general discussion of this:

Butler L. What happens when funding is linked to publication counts? In: Moed H, Glänzel W and Schmoch U (eds). Handbook of Quantitative Science and Technology Research; 2005. pp. 389–405. 

Butler L. Explaining Australia’s Increased Share of ISI Publications–the Effects of a Funding Formula Based on Publication Counts. Research Policy. 2003;32(1):143–55. doi: 10.1016/S0048-7333(02)00007-0.

Aagaard K, Bloch C, Schneider JW. Impacts of Performance-Based Research Funding Systems: The Case of the Norwegian Publication Indicator. Research Evaluation. 2015; 24(2):106–117. doi: 10.1093/reseval/rvv003.

Moed HF. UK Research Assessment Exercises: Informed Judgments on Research Quality or Quantity? Scientometrics. 2008;74(1): 153–161. doi: 10.1007/s11192-008-0108-1. 

Hammarfelt B, De Rijcke S. Accountability in Context: Effects of Research Evaluation Systems on Publication Practices, Disciplinary Norms and Individual Working Routines in the Faculty of Arts at Uppsala University. Research Evaluation. 2015;24(1):63–77. doi: 10.1093/reseval/rvu029.

Reviewer #2: Although the revised paper deals satisfactory with a few points have been raised in the original comments, it fails to address several important points.

Response: We do not aim for comprehensiveness; the added value lies mainly in comparing the data from national databases from the perspective of the perception of publication performance by evaluation systems. We do not argue with the overall relevance and importance of recommendations that we did not implement. However, we still see that some implementations (e.g. effect of changes in PRFS) are far beyond the paper's current scope and would significantly change it. We feel that addressing several recommended points would make this text on the one hand too extensive in the scope and length, on the other hand too general to address these points to a deserved extent. Therefore, we prefer to focus on the topics we set up at the beginning of the research, as stated in the article.

With full respect to the experience of Reviewer #2, we would like to thank you for these very useful recommendations, which we will be happy to address in further research projects.

---

## [Decision Letter · Decision Letter 2]

29 Mar 2021

Journal article publishing in the social sciences and humanities: a comparison of Web of Science coverage for five European countries

PONE-D-20-14121R2

Dear Dr. Petr,

We’re pleased to inform you that your manuscript has been judged scientifically suitable for publication and will be formally accepted for publication once it meets all outstanding technical requirements.

Kind regards,

Lutz Bornmann

Academic Editor

PLOS ONE

Additional Editor Comments (optional):

Reviewers' comments:

Reviewer's Responses to Questions

**Comments to the Author**

1. If the authors have adequately addressed your comments raised in a previous round of review and you feel that this manuscript is now acceptable for publication, you may indicate that here to bypass the “Comments to the Author” section, enter your conflict of interest statement in the “Confidential to Editor” section, and submit your "Accept" recommendation.

Reviewer #2: All comments have been addressed

2. Is the manuscript technically sound, and do the data support the conclusions?

Reviewer #2: (No Response)

3. Has the statistical analysis been performed appropriately and rigorously? 

Reviewer #2: (No Response)

4. Have the authors made all data underlying the findings in their manuscript fully available?

Reviewer #2: (No Response)

5. Is the manuscript presented in an intelligible fashion and written in standard English?

Reviewer #2: (No Response)

6. Review Comments to the Author

Reviewer #2: (No Response)

7. PLOS authors have the option to publish the peer review history of their article (what does this mean?). If published, this will include your full peer review and any attached files.

Reviewer #2: **Yes: **Anton J. Nederhof

---

## [Editor Report · Acceptance letter]

1 Apr 2021

PONE-D-20-14121R2 

Journal article publishing in the social sciences and humanities: a comparison of Web of Science coverage for five European countries 

Dear Dr. Petr:

I'm pleased to inform you that your manuscript has been deemed suitable for publication in PLOS ONE. Congratulations! Your manuscript is now with our production department. 

Kind regards, 

on behalf of

Dr. Lutz Bornmann 

Academic Editor

PLOS ONE